# Gintonin-Enriched Fraction Suppresses Heat Stress-Induced Inflammation through LPA Receptor

**DOI:** 10.3390/molecules25051019

**Published:** 2020-02-25

**Authors:** Sungwoo Chei, Ji-Hyeon Song, Hyun-Ji Oh, Kippeum Lee, Heegu Jin, Sun-Hye Choi, Seung-Yeol Nah, Boo-Yong Lee

**Affiliations:** 1Department of Biomedical Sciences, CHA University, Seongnam-si 13488, Gyeonggi-do, Korea; sungwoochei@gmail.com (S.C.); redcross0313@naver.com (J.-H.S.); guswl264@naver.com (H.-J.O.); joy4917@hanmail.net (K.L.); heegu94@hanmail.net (H.J.); 2Ginsentology Research Laboratory and Department of Physiology, College of Veterinary Medicine, Konkuk University, Seoul 05029, Korea; vettman@naver.com (S.-H.C.); synah@konkuk.ac.kr (S.-Y.N.)

**Keywords:** ginseng, gintonin, gintonin-enriched fraction (GEF), C2C12 cell, heat stress, inflammation

## Abstract

Heat stress can be caused by various environmental factors. When exposed to heat stress, oxidative stress and inflammatory reaction occur due to an increase of reactive oxygen species (ROS) in the body. In particular, inflammatory responses induced by heat stress are common in muscle cells, which are the most exposed to heat stress and directly affected. Gintonin-Enriched Fraction (GEF) is a non-saponin component of ginseng, a glycolipoprotein. It is known that it has excellent neuroprotective effects, therefore, we aimed to confirm the protective effect against heat stress by using GEF. C2C12 cells were exposed to high temperature stress for 1, 12 and 15 h, and the expression of signals was analyzed over time. Changes in the expression of the factors that were observed under heat stress were confirmed at the protein level. Exposure to heat stress increases phosphorylation of p38 and extracellular signal-regulated kinase (ERK) and increases expression of inflammatory factors such as NLRP3 inflammasome through lysophosphatidic acid (LPA) receptor. Activated inflammatory signals also increase the secretion of inflammatory cytokines such as interleukin 6 (IL-6) and interleukin 18 (IL-18). Also, expression of glutathione reductase (GR) and catalase related to oxidative stress is increased. However, it was confirmed that the changes due to the heat stress were suppressed by the GEF treatment. Therefore, we suggest that GEF helps to protect heat stress in muscle cell and prevent tissue damage by oxidative stress and inflammation.

## 1. Introduction

High temperature environments such as those due to global warming or hot summer seasons causes heat stress, which generates reactive oxygen species (ROS) and causes oxidative stress [1]. Oxidative stress is caused by an imbalance between the production of reactive oxygen species in the body and the ability of the biological system to remove them [2]. The redox imbalance of cells can cause toxic effects through the production of peroxides and free radicals that damage all components of the cell, including proteins, lipids, and DNA [3]. Oxidative stress caused by oxidative metabolism causes not only DNA strand breakage, but also base damage [4]. Therefore, since oxidative stress induces cytotoxicity and cell destruction, it is important that it be properly regulated.

Inflammation is an immune response by the innate immune system as a response to harmful stimuli such as pathogens and dead cells. Innate immune function can be prevented by pattern recognition receptors (PRRs) from pathogen-associated molecular pattern (PAMPs) induced by pathogen invasion and damage-associated molecular pattern (DAMPs) induced by internal stress [5]. The inflammatory response is mediated through interaction between various factors, but two major signaling pathways are known. Inflammatory response by the nuclear factor kappa-light-chain-enhancer of activated B cells (NF-κB) pathway associated with toll-like receptors (TLRs) and nucleotide-binding oligomerization domain, leucine rich repeat and pyrin domain containing 3 (NLRP3) inflammasome [6]. In addition, activation of the signaling pathway of mitogen-activated protein kinase (MAPK) induces the production of activator protein-1 (AP-1), which is transcribed in the nucleus and increases the inflammatory response [7].

Korean ginseng (*Panax ginseng* Meyer, Araliaceae) has traditionally been used as an important herb in Asia [8]. Especially, Korean red ginseng which is made by steaming and drying ginseng is known to have a positive effect on immunity enhancement, fatigue recovery, blood flow improvement, antioxidant effect, memory enhancement and menopausal disorder [9,10,11,12]. Ginseng contains various active ingredients. In particular, it contains a large amount of a saponin component which is a glycoside of steroid and triterpene [13]. Saponin of ginseng is called ginsenoside and has been studied to exert various effects on body function by affecting the central nervous system, endocrine system, immune system, metabolism system [13,14,15,16]. However, ginseng contains various components not only in the saponin portion but also in the non-saponin portions, but the effects of these components are unclear. Gintonin-enriched fraction (GEF) is a non-saponin ingredient, which has been attracting attention in recent years. GEF is based on signal transduction through the lysophosphatidic acid (LPA) receptors. GEF is composed of a large amount of lysophosphatidic acids as a functional group and contains a large amount of linoleic acid and phosphatidic acids, as lipid components [17]. Therefore, we investigated the effects of GEF, one of the non-saponin components, on oxidative stress and inflammation induced by heat stress and prove the molecular mechanisms.

## 2. Results

### 2.1. Effect of GEF on Cell Viability and Expression of MAPK Signaling Factors Induce by Heat Stress

A thiazolyl blue tetrazolium bromide (MTT) assay was performed to determine the proper concentration of GEF not showing toxicity in cells. Measurements showed no toxicity up to 100 μg/mL (Figure 1b). Thereafter, the experiment was carried out using two concentrations (20 and 100 μg/mL) in the range of not showing toxicity and the appropriate heat stress temperature revealed in the existing studies [12]. To determine whether heat stress affects the MAPK signaling pathway in C2C12 cells, phosphorylation of p38 and ERK was confirmed in cells exposed to heat stress for 1 h. As shown in Figure 1c, the expression of phosphorylated p38 and ERK was increased by heat stress but decreased by GEF treatment. In addition, the band density was significantly different from that of total p38 and ERK protein (Figure 1d).

### 2.2. Effect of GEF on Oxidative Stress and Inflammatory Response by Heat Stress

The expression of antioxidant enzymes was confirmed in order to determine whether GEF inhibits oxidative stress caused by heat stress. As shown in Figure 2a, when exposed to heat stress, glutathione peroxidase (GPx) and glutathione reductase (GR) expression increased in order to eliminate increased reactive oxygen species (ROS). However, GPx and GR were not increased by suppressing the production of ROS induced by heat stress when GEF was treated. This decrease in protein expression was significantly different in the quantification graph (Figure 2b).

In addition, the expression of HO-1, which plays a role in inhibiting the inflammatory reaction in the presence of excessive ROS production, inflammation, and high temperature, was confirmed. As a result, it was confirmed that HO-1 increased by heat stress was further decreased by GEF treatment than control. The expression of Nuclear factor erythroid-2-related factor 2 (Nrf2), which protects against oxidative damage and results in the induction of many cytoprotective proteins, was confirmed. The expression level of Nrf2 was increased to protect against oxidative damage caused by heat stress and reduced because GEF protects against oxidative damage. We also investigated Nitric Oxidative (NO) production using the Griess reagent, since the ROS play an important role and involved in regulation of the biologically effective concentration of NO. Heat stress induced NO production suppressed by pretreated with GEF (Figure 2c).

We investigated whether heat stress induced inflammation in muscle cells and confirmed the expression of inflammatory factors in order to confirm the effect of GEF. We confirmed that protein expression of NLRP3 inflammasome, another pro-inflammatory cytokine production pathway. Expression of NLRP3, Apoptosis-associated speck-like protein containing a CARD (ASC) and caspase-1, which are major components of NLRP3 inflammasome, were excessively increased by heat stress but decreased dramatically by GEF treatment (Figure 2d). These results, as well as the above results, show significant differences through the quantification graph.

### 2.3. Effect of GEF on Expression of Inflammatory Cytokines by Heat Stress

To determine inflammatory signals induced by heat stress lead to secretion of inflammatory cytokines, protein expression of inflammatory cytokines was confirmed in cells exposed to heat stress for 15 h. As shown in Figure 3, when exposed to heat stress, expression of mature forms of IL-18 and IL-1β changed by the NLRP3 inflammasome, were increased. However, increment of expression was suppressed to a level similar to that of control by GEF treatment. In addition, IL-6 and COX-2 were showed similar patterns of expression.

### 2.4. GEF Affects MAPK Signaling Pathway and Inflammatory Cytokines Through LPA Receptor

GEF effects on MAPK signaling pathway and interleukins through LPA receptor. In previous experiments, we confirmed that GEF inhibits inflammatory responses through the MAPK signaling pathways. We used the LPA receptor antagonist (ki16425A), assuming that GEF binds to the LPA receptor. The expression of MAPK-related factors was identified at 1-h heat stress. Phosphorylated forms of p38 and ERK by heat stress were significantly increased compared to the controls (Figure 4a). And the inflammatory response was suppressed in the expression level of the GEF-treated group. When antagonist was treated only, the expression level was not different from inflammatory response by heat stress but when GEF and antagonist were treated together, the expression level of inflammation was higher than that of GEF-treated group. The expression of IL-18, IL-1b, IL-6 and COX-2 by 15 h heat stress was also decreased by GEF treatment (Figure 4b). The combination of GEF and antagonists increased the inflammatory response compared to the GEF-treated group. These suggested that GEF affects MAPK signaling through the LPA receptor, which ultimately inhibits inflammatory responses such as Interleukins and COX-2.

## 3. Discussion

Heat stress causes oxidative stress and inflammatory reactions, causing extensive damage to the body and, in severe cases, death [18]. Therefore, in order to suppress oxidative stress and inflammatory reaction due to heat stress, the effect of C2C12 cells was confirmed by using GEF. The muscle that occupies the most weight in the body is most affected by the heat stress. Thus, we used muscle cells to confirm the protective effect against heat stress. The results showed that GEF effectively controlled both oxidative stress and inflammatory responses induced by heat stress. GEF was shown to affect the inflammatory response through regulation of p38 and ERK in the early part of the heat stress exposure. After that, in cells exposed to heat stress for 12 h, it has also been shown to affect the activation of NLRP3 inflammasome. In particular, NLRP3 inflammasome is composed of three constitutive proteins: NLRP3, ASC, and caspase-1. GEF regulates all three proteins, suggesting that GEF effectively inhibits the inflammatory response through NLPR3 inflammasome. These results may be confirmed by the secretion of inflammatory cytokines. The expression of IL-18 and IL-1beta, a cytokine that is converted to mature form by NLRP3 inflammasome, is reduced by GEF [19]. The expression of antioxidant enzymes such as catalase, GR, and HO-1 were increased in order to protect the body damage due to oxidative stress caused by ROS when exposed to heat stress. However, it is considered that the expression of antioxidant enzymes is decreased because the production of ROS is reduced by GEF treatment. These results clearly demonstrate the effect of GEF treatment in a dose dependent manner. In particular, HO-1 is known to be increased by heat in many previous studies and is known to be associated with the expression of heat shock protein (HSP) [20,21]. Our previous studies have also shown that HSP expression is increased by environmental heat stress [9]. These results suggest that GEF may also affect the expression of HSP and HSP associated signaling pathways. In addition, the expression of Nrf2, which is an upstream regulator of HO-1, is also similar to the expression of HO-1, suggesting that GEF is effective in the regulation of the HO-1 signaling pathway. Interestingly, GEF contains a lot of LPA components and LPA is known to cause anti-inflammation according to previous studies [22,23,24]. However, GEF dramatically inhibited oxidative stress and inflammation induced by heat stress, and other studies have shown that it inhibits inflammation [25,26]. Through the study, we have shown that GEF, a non-saponin portion of ginseng, has potential as a good functional material to prevent heat stress. We found that GEF inhibits the inflammatory response increased by heat stress. We identified which pathway inhibits the inflammatory response. Previous studies have shown that GEF contains several types of LPA, an endogenous phospholipid-derived growth factor in animals [27]. There are several receptors in the cell membrane that bind GEF, but we hypothesized that the LPA receptor would be the main receptor. LPA enters the cell membrane by the LPA receptor and activates the G protein in the cell. The activated G protein induces an inflammatory response through the MAPK signaling pathway [28]. Therefore, LPA receptor antagonist was used to examine the inflammatory signaling pathway of GEF. Inflammatory response by heat stress was decreased on GEF only treated group. When antagonist and GEF were treated together, the expression level of inflammatory response was higher than the GEF-treated only group. We confirmed that GEF effects on the LPA receptor. Inflammatory response was reduced in the GEF and antagonist-treated group at 1hour heat stress, indicating that GEF binds to LPA receptor as main receptor. IL-18, IL-1β IL-6 and COX-2 expressed at 15hour heat stress were the same as the expression level of antagonist-treated group or increased in GEF-treated only group.

In conclusion, GEF inhibited increased inflammatory response by heat stress through NLRP3 inflammasome and MAPKs signaling pathway. In addition, we have found that GEF suppressed the inflammation by heat stress through LPA receptor.

## 4. Materials and Methods

### 4.1. Preparation of Gintonin-Enriched Fraction (GEF)

Gintonin-enriched fraction prepared by applying a previously described method [17,29]. In summary, one kg of 4-year-old ginseng were ground into small pieces (>3 mm) and refluxed with 70% edible Ethanol (EtOH) for 8 h at 80 °C for eight times. The EtOH extracts were concentrated (350 g) and dissolved in distilled cold water at a ratio of 1–10 and stored in a cold chamber at 4 °C for 24 h. After centrifugation, the precipitate was freeze dried. This fraction was designated GEF. The yield of GEF is 1.3%. The total proteins were approximately 30.3%, carbohydrates were 30% and lipid was 20.2% in GEF [30].

### 4.2. Chemical and Reagent

Antibodies against glutathione reductase (GR), glutathione peroxidase (GPx), heme oxygenase 1 (HO-1) and glyceraldehyde 3-phosphate dehydrogenase (GAPDH) were purchased from Santa Cruz Biotechnology (Dallas, TX, USA). Antibodies against NLRP3, caspase-1, p-p38 and phosphorylated extracellular signal-regulated kinases (p-ERK) were purchased from Cell Signaling Technology (Beverly, MA, USA). Thiazolyl blue tetrazolium bromide (MTT) was purchased from Alfa Aesar Chemical Inc. (Ward Hill, MA, USA). Unless noted otherwise, all chemicals were purchased from Sigma Chemical Co. (St. Louis, MO, USA). Ki16425 were purchased from Cayman Chemicals (Ann Arbor, MI, USA).

### 4.3. Cell Study Design

C2C12 cells were purchased from the American Type Culture Collection (Manassas, VA, USA). Cells were cultured in Dulbecco’s Modified Eagle’s Medium (DMEM) containing 10% heat-inactivated fetal bovine serum (FBS, Invitrogen, Carlsbad, CA, USA) and 100 mg/mL penicillin-streptomycin and grown at 37 °C in a 5% CO_2_ air environment. C2C12 cells were seeded at a density of 1 × 104 cells per well in 96-well plates. The MTT assay was performed as described previously [9]. To obtain results of MTT assays and ELISA kits, absorbance was measured at 570 nm on a PowerWave HT ELISA reader (BioTek, Winooski, VT, USA). As shown in (Figure 1a), for the cell-based heat stress model, C2C12 cells were seeded in 100 mm dishes at a density of 1 × 106 cells/dish. After overnight, the cells were treated with GEF for 4 h, and then exposed to heat stress (43 °C) for 1h or 12 h. After heat exposure period, the cells were harvested.

### 4.4. Western Blotting

C2C12 cells were washed with PBS, homogenized in 50 μL of radio immunoprecipitation assay (RIPA) buffer (50 mM Tris-HCl (pH 7.4), 150 mM NaCl, 1 mM ethylenediaminetetraacetic acid (EDTA), 1% Triton X-100, 1% sodium deoxycholate, and 0.1% SDS) supplemented with protease inhibitors (1 mM phenylmethylsulfonyl fluoride (PMSF), 5 mg/mL aprotinin, and 5 mg/mL leupeptin) and phosphatase inhibitors (1 mM Na3VO4 and 1 mM NaF), and centrifuged at 15,500 g for 10 min at 4 °C. The protein concentration was determined by the bicinchoninic acid (BCA) assay (Pierce, Rockford, IL, USA) using bovine serum albumin as the standard. Proteins (20 mg) were separated by sodium dodecyl sulfateepolyacrylamide gel electrophoresis (SDS-PAGE) and transferred to nitrocellulose membranes (Osmonics, Minnetonka, MN, USA). Membranes were incubated with a specific primary antiserum in tris-buffered saline containing 0.05% Tween-20 and 5% nonfat dry milk overnight at 4 °C. After three washes with trisbuffered saline containing 0.05% Tween-20, membranes were incubated with peroxidase-conjugated IgG for 1 h at room temperature, visualized using enhanced chemiluminescence (Amersham Biosciences, Piscataway, NJ, USA), and photographed using the ChemiDoc system and Image Lab Software (Bio-Rad Laboratories, Hercules, CA, USA) [31].

### 4.5. Nitric Oxidative (NO) Assay

C2C12 cell pretreated with GEF (0, 20, 100 µg/mL) for 4 h, and stimulated with heat for 12 h. The culture supernatants were collected and mixed 100 µL Griess reagent (Merck Millipore, Burlington, MA, USA). After 15 min, the level of nitric in culture supernatant was measured by ELISA in 570 nm [32].

### 4.6. Statistical Analysis

Differences among multiple groups were determined by one-way analysis of variance (ANOVA), followed by Duncan’s multiple range test, using the SPSS software system (SPSS for Windows, version 20; SPSS, Inc., Chicago, IL, USA). Values with different letters are significantly different, *p* < 0.05.

## Figures and Tables

**Figure 1 molecules-25-01019-f001:**
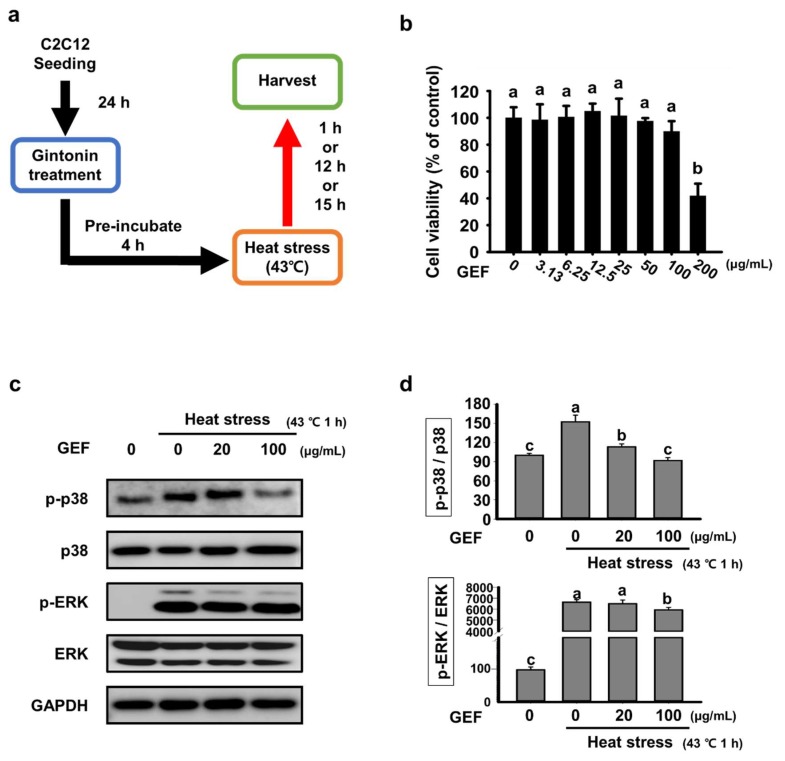
Effects of gintonin-enriched fraction (GEF) on cell viability and expression of MAPK signaling factors in C2C12 cells under heat-exposed conditions (**a**) Cells were seeded and treated with GEF (20 or 100 μg/mL). After 4 h of pre-incubation, the cells were incubated at 43 °C to induce heat stress; (**b**) Cell viability was measured by thiazolyl blue tetrazolium bromide (MTT) assay in a dose-dependent manner; (**c**) Analysis of p-p38 and p-ERK expression by western blot; (**d**) The density of p-p38 and p-ERK was quantified and expressed as a bar graph. Values labeled with different letters are significantly different (*p* < 0.05).

**Figure 2 molecules-25-01019-f002:**
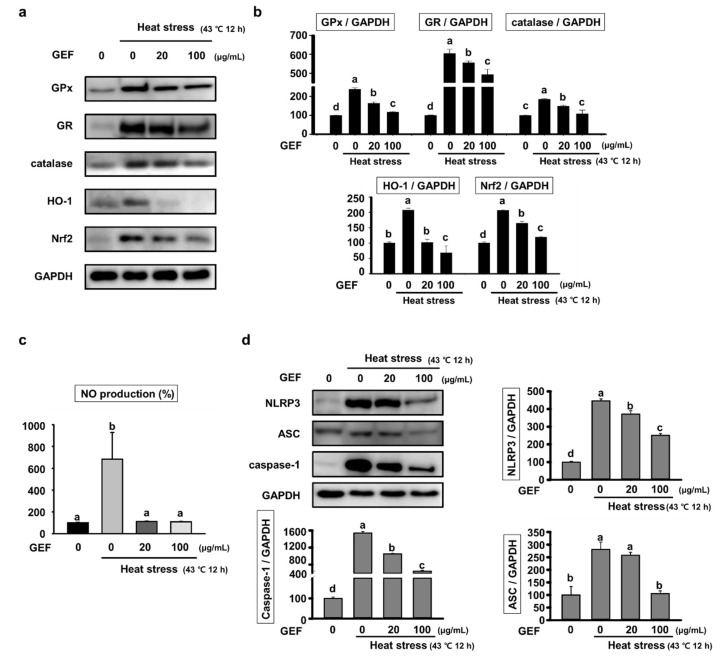
Expression of oxidative stress–related proteins and NLRP3 inflammasome in C2C12 cells under heat-exposed conditions; (**a**) analysis of glutathione peroxidase (GPx), glutathione reductase (GR), catalase, HO-1 and Nuclear factor erythroid-2-related factor 2 (Nrf2) expression by western blot; (**b**) the density of GPx, GR, catalase, HO-1 and Nrf2 versus glyceraldehyde 3-phosphate dehydrogenase (GAPDH) was quantified and expressed as a bar graph, respectively; (**c**) the level of NO in the culture media in C2C12 cells pretreated with GEF (0, 20, 100 µg/mL) for 4 h and heat stress for 12 h was estimated using Griess reagent; (**d**) Expression of proteins related to inflammation, including NLRP3, Apoptosis-associated speck-like protein containing a CARD (ASC) and caspase-1 analyzed by Western blot. The density of NLRP3, ASC and caspase-1 versus GAPDH was quantified and expressed as a bar graph, respectively. Values labeled with different letters are significantly different (*p* < 0.05).

**Figure 3 molecules-25-01019-f003:**
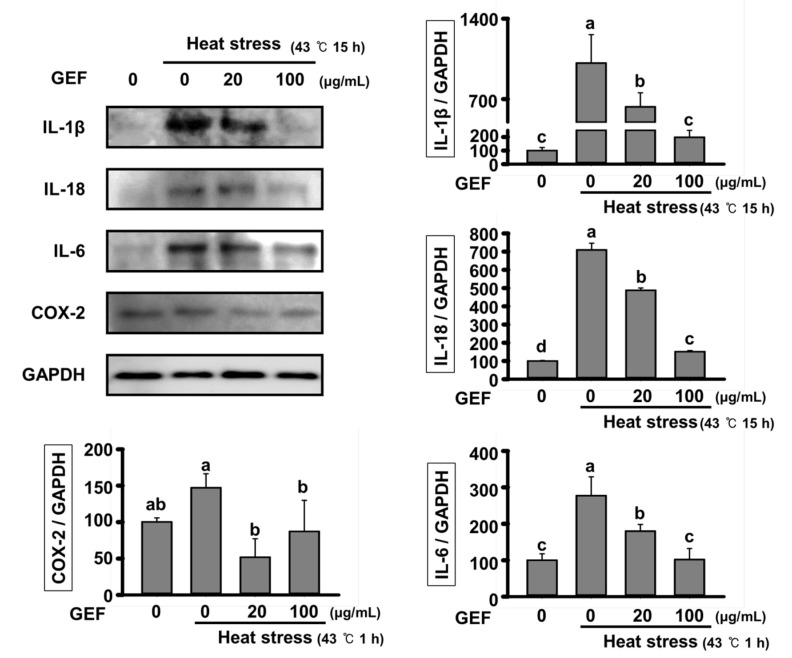
Expression of inflammatory cytokines in C2C12 cells under heat-exposed conditions Expression of interleukin (IL)-18, IL-1β, IL-6 and COX-2 analyzed by Western blot. The density of IL-18, IL-1β, IL-6 and COX-2 versus GAPDH was quantified and expressed as a bar graph, respectively. Values labeled with different letters are significantly different (*p* < 0.05).

**Figure 4 molecules-25-01019-f004:**
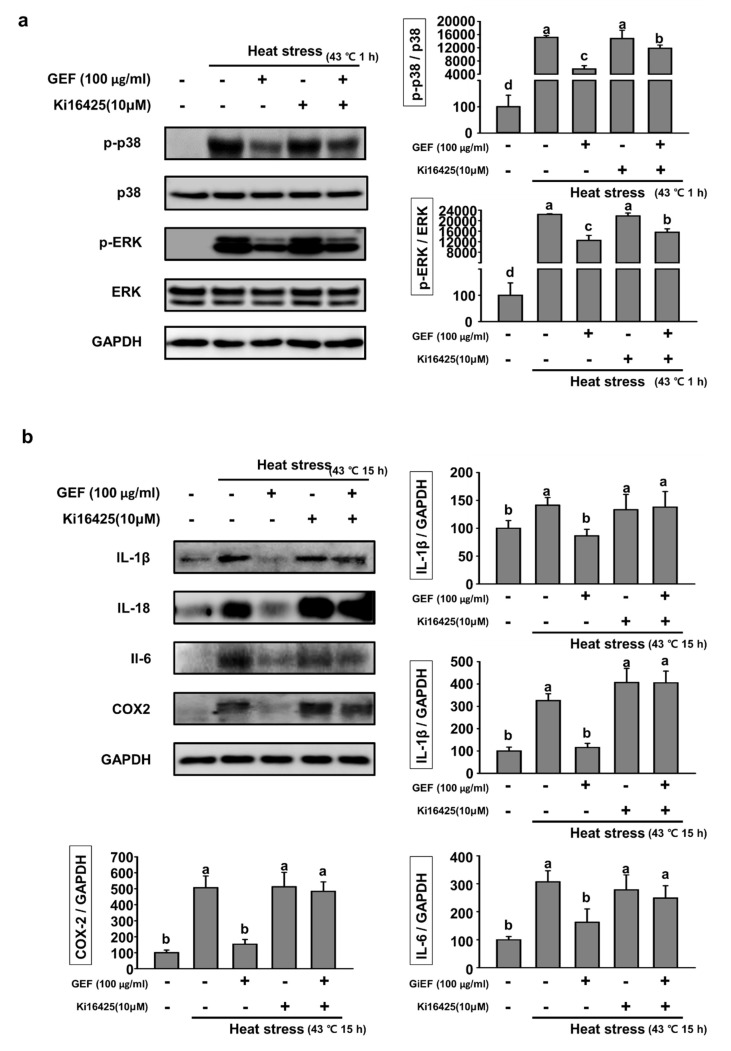
Expression of MAPK signaling factors and inflammatory cytokines with antagonist and GEF on heat stress (**a**) Analysis of antagonist (Ki16425) on p-p38 and p-ERK expression by Western blot. Quantification of p-p38 per total p38 was represented by a bar graph and quantification of p-ERK per total ERK was represented by a bar graph. (**b**) Expression of inflammatory cytokines treated with antagonist and GEF at heat stress. Analysis of the effect of antagonist on IL-18, IL-1β, IL-6 and COX-2 expression by Western Blot. Quantification of IL-18, IL-1β, IL-6 and COX-2 per GAPDH was represented by a bar graph, respectively. Values labeled with different letters are significantly different (*p* < 0.05).

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
