# Peer review of "Gintonin-Enriched Fraction Suppresses Heat Stress-Induced Inflammation through LPA Receptor"

_molecules, 2020, doi:10.3390/molecules25051019_

Round 1
Reviewer 1 Report
Heat stress causes oxidative stress and inflammation in the muscles. Therefore, the authors have evaluated the in vitro effect of Gintonin-Enriched Fraction (GEF) (non-saponin component of ginseng) against heat stress-induced Inflammation. C2C12 cells were exposed to high temperature in the presence or absence of GEF. and various markers of oxidative stress and inflammation were measured. GEF protected against heat-stress induced toxicity.
Figures are not clear. Please do it properly. What is a,b,c?
Rather than using the word “muscle tissue” the author can use the word “muscle”
Please specify the type of muscle
Author Response
"Please see the attachment"
Reviewer 1
Heat stress causes oxidative stress and inflammation in the muscles. Therefore, the authors have evaluated the in vitro effect of Gintonin-Enriched Fraction (GEF) (non-saponin component of ginseng) against heat stress-induced Inflammation. C2C12 cells were exposed to high temperature in the presence or absence of GEF. and various markers of oxidative stress and inflammation were measured. GEF protected against heat-stress induced toxicity.
- Thanks for your kind comment. We are very pleasure to have been given the opportunity to revise our manuscript. We carefully revised our manuscript according to your comments.
Figures are not clear. Please do it properly. What is a,b,c?
- We carefully revised our manuscript point-by-point and made the necessary changes to the manuscript. We changed to proper indicated as a, b, c in Figure 1 and 2.
Rather than using the word “muscle tissue” the author can use the word “muscle” Please specify the type of muscle
- We revised the word “muscle tissue” to “muscle cell” in line 16 and 27.

Reviewer 2 Report
In this paper, Chei et al described effects go Gintonin enriched fraction on heat stress induced cytokine and implication of LPA receptor.
Nevertheless I think that the results are too preliminary to be published in molecules and I have some issues about the experimental design and presented results.
The authors used giintonin-enriched fraction extract from ginseng. As they mentioned in M&M section, they have a laboratory routine to extract this fraction. Nevertheless, what compounds are really into this fraction. How to manage the purity and the concentration of bio-active compounds in each extract. For this paper, how many extracts are used and how the authors normalized concentration of bioactive compounds. Extracts were made by multiple researchers. Are they always obtained the same biological activity. Moreover, the authors said that GEF inhibits oxidative stress but they measured oxidative stress indirectly by reporting Gpx and Gr. The authors have to measure ROS produced during heat stress. In figure 3 the authors measured production of inflammatory cytokines by western blot. Western blot is not appropriate method to measure cytokine production as it is not quantitative and we can not be sure that the cytokine are secreted into culture media. So the authors have to replace western blot by ELISA quantification. In figure 4 the authors want to show that LPA receptor is involved in GEF effects using KI16425 which is a pharmacological agonist. When we perform this kind of study we prefer to use agonists and antagonists compounds, and the authors have to show by competition that GEF replace agonist from LPA. Moreover describing map-kinase pathway is not sufficient and probably LPA receptor implicated other downstream signaling like pi3 kinase nf-kappa and so on As they showed GEF decreased pERk and pP38 with GEF to block the induced inflammatory processus. p38 and ERK inhibitors also do it ? In figure 4 cytokines productions have to be measure by ELISA.Author Response
"Please see the attachment"
Reviewer 2
In this paper, Chei et al described effects go Gintonin enriched fraction on heat stress induced cytokine and implication of LPA receptor. Nevertheless I think that the results are too preliminary to be published in molecules and I have some issues about the experimental design and presented results.
- Thanks for your kind comment. We are very pleasure to have been given the opportunity to revise our manuscript. We carefully revised our manuscript according to your comments.
The authors used gintonin-enriched fraction extract from ginseng. As they mentioned in M&M section, they have a laboratory routine to extract this fraction. Nevertheless, what compounds are really into this fraction. How to manage the purity and the concentration of bio-active compounds in each extract. For this paper, how many extracts are used and how the authors normalized concentration of bioactive compounds. Extracts were made by multiple researchers. Are they always obtained the same biological activity.
- Thank you for your proper comment. We used Gintonin from Dr.Nah’s laboratory. Although this gintonin is a laboratory routine to extract, this material has been studied for a long time, and it is verified by a lot of peer reviewed paper [1-5]
Moreover, the authors said that GEF inhibits oxidative stress but they measured oxidative stress indirectly by reporting Gpx and Gr. The authors have to measure ROS produced during heat stress. In figure 3 the authors measured production of inflammatory cytokines by western blot. Western blot is not appropriate method to measure cytokine production as it is not quantitative and we can not be sure that the cytokine are secreted into culture media. So the authors have to replace western blot by ELISA quantification.
- We added NO production to provide more evidence the effect of GEF in ROS (Figure 2C), and we also added result part (line 97-100) and materials & methods part (line 245-248). We also agree with ELISA quantification is more proper method to measure the cytokine production. Unfortunately, we were not in the situation to performed the ELISA experiment, so we had to determine the cytokine production by western blot.
In figure 4 the authors want to show that LPA receptor is involved in GEF effects using KI16425 which is a pharmacological agonist. When we perform this kind of study we prefer to use agonists and antagonists compounds, and the authors have to show by competition that GEF replace agonist from LPA. Moreover describing map-kinase pathway is not sufficient and probably LPA receptor implicated other downstream signaling like pi3 kinase nf-kappa and so on As they showed GEF decreased pERk and pP38 with GEF to block the induced inflammatory processus. p38 and ERK inhibitors also do it? In figure 4 cytokines productions have to be measure by ELISA.
- Thank you for your kind comment. We would like to determine that LPA receptor is involved in GEF effect using LPA antagonist, KI16425. As your advice, experimenting with both agonist and antagonist compound may prove more clearly, however, we only determined using antagonist, KI16425 due to the financial problem.
- Our data seems not to be the only LPA downstream signaling in p-ERK and p-p38, however we determined that LPA signaling is PRIMARY pathway with GEF since the LPA antagonist with GEF treated group increased than GEF treated group in heat stress stimulated. In addition, we also going to determine another pathway LPA receptor with GEF treatment in further study
References
- Sun-Hye Choi, Seok-Won Jung, Byung-Hwan Lee, Hyeon-Joong Kim, Sung-Hee Hwang, Ho-Kyoung Kim and Seung-Yeol Nah: Ginseng pharmacology: a new paradigm based on gintonin-lysophosphatidic acid receptor interactions. Pharmacol, 27 October 2015
- Choi JH, Jang M, Oh S, Nah S-Y, Cho I-H: Multi-target protective effects of gintonin in 1-methyl-4-phenyl-1, 2, 3, 6-tetrahydropyridine-mediated model of Parkinson's disease via lysophosphatidic acid receptors. Frontiers in pharmacology 2018, 9:515
- Choi S-H, Shin T-J, Lee B-H, Hwang SH, Kang J, Kim H-J, Park C-W, Nah S-Y: An edible gintonin preparation from ginseng. Journal of ginseng research 2011, 35:471.
- Choi S-H, Jung S-W, Kim H-S, Lee B-H. Kim J-Y, Hwang S-H, Rhim H, Kim H-C, Nah S-Y: A brief method for preparation of gintonin-enriched fraction from ginseng. Journal of ginseng research 2015, 398:405.
- Choi JH, Jang M, Oh S, Nah S-Y, Cho I-H: Multi-target protective effects of gintonin in 1-methyl-4-phenyl-1, 2, 3, 6-tetrahydropyridine-mediated model of Parkinson's disease via lysophosphatidic acid receptors. Frontiers in pharmacology 2018, 9:515

Reviewer 3 Report
The manuscript investigated protective effect of GEF on heat stress-induced cellular inflammation and damage and involvement of LPA receptor. Overall, the manuscript is easy to follow but the manuscript did not provide any information regarding active compounds in GEF fraction. As the authors indicated, GEF is a mixture of diverse phytochemicals and synergistic effects among phytochemicals can be involved in the suppression of heat stress-induced cellular inflammation.
Major concerns
The characterization of GEF and concentration of various phytochemicals in GEF should be indicated. The reduction of cellular apoptosis and ROS production by GEF can be more direct evidence against heat-stress. Please provide data if you have some of them. Source of ginseng used for GEF preparation should be indicated. Resolution of Figure 3 need to be improved.Minors
76 Figurer: correct to FigureThe extent of relative p-ERK expression is unusual. 70-fold increase in protein expression by heat stress?
164 ho-1: correct to HO-1Author Response
"Please see the attachment"
Reviewer 3
The manuscript investigated protective effect of GEF on heat stress-induced cellular inflammation and damage and involvement of LPA receptor. Overall, the manuscript is easy to follow but the manuscript did not provide any information regarding active compounds in GEF fraction. As the authors indicated, GEF is a mixture of diverse phytochemicals and synergistic effects among phytochemicals can be involved in the suppression of heat stress-induced cellular inflammation.
- Thanks for your kind comment. We are very pleasure to have been given the opportunity to revise our manuscript. We carefully revised our manuscript according to your comments.
Major concerns
The characterization of GEF and concentration of various phytochemicals in GEF should be indicated. The reduction of cellular apoptosis and ROS production by GEF can be more direct evidence against heat-stress. Please provide data if you have some of them. Source of ginseng used for GEF preparation should be indicated. Resolution of Figure 3 need to be improved.
- We added NO production to provide more evidence the effect of GEF in ROS (Figure 2C), and we also added result part (line 97-100) and materials & methods part (line 245-248).
- We also improved the resolution of Figure 3.
Minors
76 Figurer: correct to Figure
- We revised the word “Figurer” to “Figure” in line 77.
The extent of relative p-ERK expression is unusual. 70-fold increase in protein expression by heat stress?
- Since this Western blot data was quantified through “Image J” program, there seems to be a lot of difference.
164 ho-1: correct to HO-1
- We revised the word “ho-1” to “HO-1” in line 167.

Round 2
Reviewer 2 Report
I would thank the authors which attempt to complete their data. I'm still not convinced by the demonstration, NO measurement is not exactly reliable to ROS production within the cells. These molecules neither have the same signaling pathways nor the same kinetics within the cells. Moreover it s not sure that detoxifying enzymes are the same. There are some other tests to measure ROS into cells and authors have to use them.
I know that cytokine expression may be assessed by western blot but it's not quantitative. Furthermore, we are not sure that cytokines are secreted. The authors have to measure cytokine by ELISA which it's a gold standard methods if they want to claim that gitonin fractions suppresses inflammatory.
Elisa can be done by our own, it not necessary to buy kits, we need 96 plates, recombinant cytokines for the standard, capture antibodies and some other reagents. There are lot of methods papers see for instance : Chiswick EL, Duffy E, Japp B, Remick D. Detection and quantification of cytokines and other biomarkers. Methods Mol Biol. 2012;844:15–30. doi:10.1007/978-1-61779-527-5_2.
Author Response
Reviewer 2
I would thank the authors which attempt to complete their data. I'm still not convinced by the demonstration, NO measurement is not exactly reliable to ROS production within the cells. These molecules neither have the same signaling pathways nor the same kinetics within the cells. Moreover it’s not sure that detoxifying enzymes are the same. There are some other tests to measure ROS into cells and authors have to use them.
- We are very appreciated for your careful review. The expression of GPx, Gr, catalase, and NO production occurs right after the ROS production. GPx, Gr, catalase, and NO are downstream genes in ROS generation and elimination pathway and also well known that expression of these genes occurs after ROS production [1]. Our experiment more focused these downstream genes to explain more clearly the ROS production.
I know that cytokine expression may be assessed by western blot but it's not quantitative. Furthermore, we are not sure that cytokines are secreted. The authors have to measure cytokine by ELISA which it's a gold standard methods if they want to claim that gitonin fractions suppresses inflammatory.
Elisa can be done by our own, it not necessary to buy kits, we need 96 plates, recombinant cytokines for the standard, capture antibodies and some other reagents. There are lot of methods papers see for instance : Chiswick EL, Duffy E, Japp B, Remick D. Detection and quantification of cytokines and other biomarkers. Methods Mol Biol. 2012;844:15–30. doi:10.1007/978-1-61779-527-5_2.
- I completely agree with your comment. Unfortunately, we do not have any cell supernatant left to measure the cytokine, since we used the last few samples to measure NO production. We are now in a difficult situation to restart the experiment from scratch. In addition, Generally the researchers use both Western blot and ELISA to measure the cytokine, but the other researchers are measured cytokines by Western blot only. I think that measuring the cytokine by western blot only is not just inaccurate method. Aaccording to the Wang et al. in Molecules 2019, cytokine measured by western blot only [2].
[References]
- Wang, K.; Zhang, T.; Dong, Q.; Nice, E.C.; Huang, C.; Wei, Y. Redox homeostasis: the linchpin in stem cell self-renewal and differentiation. Cell Death Dis 2013, 4, e537, doi:10.1038/cddis.2013.50.
- Wang, R.; Dong, Z.; Lan, X.; Liao, Z.; Chen, M. Sweroside Alleviated LPS-Induced Inflammation via SIRT1 Mediating NF-kappaB and FOXO1 Signaling Pathways in RAW264.7 Cells. Molecules 2019, 24, doi:10.3390/molecules24050872.